# Influenza B: Prospects for the Development of Cross-Protective Vaccines

**DOI:** 10.3390/v14061323

**Published:** 2022-06-17

**Authors:** Liudmila M. Tsybalova, Liudmila A. Stepanova, Edward S. Ramsay, Andrey V. Vasin

**Affiliations:** 1Smorodintsev Research Institute of Influenza, Prof. Popova Str., 15/17, 197376 St. Petersburg, Russia; stepanoval60@mail.ru (L.A.S.); warmsunnyday@mail.ru (E.S.R.); or office@spbstu.ru (A.V.V.); 2Research Institute of Influenza named after A.A. Smorodintsev, Peter the Great St. Petersburg Polytechnic University, Polytechnicheskaya, 29, 195251 St. Petersburg, Russia

**Keywords:** influenza B, virus, phylogenetic lineages, viral proteins, monoclonal antibodies, cross-protection vaccines

## Abstract

In this review, we analyze the epidemiological and ecological features of influenza B, one of the most common and severe respiratory infections. The review presents various strategies for cross-protective influenza B vaccine development, including recombinant viruses, virus-like particles, and recombinant proteins. We provide an overview of viral proteins as cross-protective vaccine targets, along with other updated broadly protective vaccine strategies. The importance of developing such vaccines lies not only in influenza B prevention, but also in the very attractive prospect of eradicating the influenza B virus in the human population.

## 1. Background

Influenza B is one of the most common respiratory infections, causing a considerable public health burden [1,2]. A number of studies have shown that 20–30% of diagnosed influenza viral infections are due to influenza B viruses [3,4]. An analysis of the Global Influenza B Study (GIBS) database, covering a total of 1,820,301 influenza cases (from 2000 to 2018) in 31 countries, showed that influenza B accounted for 419,167 (23.4%) of the cases [1]. During some influenza epidemics, diseases associated with influenza B are registered with the same frequency, or even more often, than influenza A/H1N1 or A/H3N2. Influenza B viruses are important contributors to the morbidity and mortality associated with influenza epidemics [3].

Simultaneous circulation of two antigenically different phylogenetic lineages B/Yamagata and B/Victoria and two clades of B/Yamagata lineage complicate the choice of vaccine strains and demonstrate the need to develop a broadly protective vaccine. Vaccine strain selection is also complicated by the relative endemicity of influenza B viruses, as a result of which the vaccine strains recommended by the WHO for the entire hemisphere may not match the viruses circulating in a particular region.

One of the most urgent tasks declared by the WHO for the next decade in the “WHO preferred product characteristics for next generation influenza vaccines” is to create vaccines that provide long-term protection against severe and complicated forms of influenza A and B. The development of such universal vaccines is underway in many countries, but the main focus is on influenza A vaccines. In this article, we have attempted to provide an overview of the current developments of cross-protection influenza B vaccines.

Several conserved viral antigens of HA stalk, NA, and T cell epitopes in the internal proteins such as PB1, NP, and M1 can be the basis for the development of such a broadly protective vaccine against influenza B. Studying the mAb repertoire will help identify target proteins/peptides and guide the design of novel IBV vaccines for broader protection.

## 2. Epidemiological and Ecological Features of Influenza B Infection

Influenza B viruses (FLUBVs) usually co-circulate with influenza A viruses (FLUAVs); and they are involved as etiological agents in almost all annual influenza epidemics. In some epidemics, they are predominant [3,4,5,6]. Similar data were obtained at the WHO National Influenza Centre of Russia (Smorodintsev Research Institute of Influenza, St. Petersburg). During three epidemic seasons of the last decade (2012–2013, 2014–2015, 2016–2017), influenza B-associated illnesses were registered with the same frequency as influenza A (AH1N1, AH3N2). In the 2017–2018 epidemic, influenza B-associated illnesses were twofold higher than illness caused by influenza A (unpublished data.) According to the GIBS database, the influenza B virus has been the dominant virus type in about one in every seven seasons [1]. 

The proportion of influenza B cases among overall influenza cases varies geographically. Its median value is significantly higher (*p* = 0.060) in countries of the inter-tropical belt 27.4% (IQR 12.2–41.7%), compared to temperate countries of the Northern 21.0% (IQR 7.3–37.4%) or Southern 22.2% (IQR 9.1–34.5%) hemispheres [1]. The majority of IBV infections occurred in children and adolescents [4,7,8,9,10,11,12]. It is widely believed that influenza B is a less severe infection than influenza A, but current studies challenge this notion. There is no difference between influenza A and influenza B in the frequency of hospitalization, intensive care unit (ICU) admission, or rate of death among hospitalized influenza patients [13]. Moreover, influenza B has been described as a more severe illness in younger age groups and those aged 60 years or more [11,14,15,16].

The 2017–2018 season with dominant circulation of influenza B featured the highest number of severe cases. According to information on European Union/European Economic Area countries, among all of the cases reported with the influenza B infection in ICUs and a known outcome, 420/1983 (21%) died in 2017/18. [14]. Among fatal influenza B cases, 79% (333/420) were 60 years of age or older. A similar correlation between virus type and age was also observed in the B/Yamagata virus-dominated 2012–2013 season [14]. During the 2019–2020 influenza season, the highest hospitalization rate due to influenza B infection was among adults 65 years or older [17]. The proportion of deaths associated with pediatric influenza B infection was greater than that of influenza A among children age ≥10 years (relative to <6 months) and was associated with the greatest odds of ICU admission [18]. An analysis of influenza surveillance data for the period 2004–2019 in the U.S. revealed that a range (7–51%) of pediatric deaths with confirmed influenza were associated with influenza B, excluding 2009–2010 pandemic mortality [17].

In the 2012–2013 season, the level of pediatric deaths attributed to influenza B in the U.S. reached 52% of fatal pediatric influenza cases [19]. Due to the emergence of a second, antigenically distinct FLUBV lineage, as well as social factors (changing demographics, increased urbanization, increased mobility of the global population), FLUBV evolution and spread are accelerating [3,20]. Unlike influenza A viruses, evidence sustained animal reservoir for influenza B viruses has not been found. Influenza B isolation has only been reported in seals (Figure 1) [21]. It has also been shown that pigs are susceptible to influenza B viruses. A serological survey of domestic swine herds showed that antibodies to influenza B viruses were detected in 7.3% of the tested swine serum [22]. Further, influenza B viruses was detected by RT-PCR in three nasal swabs collected from swine with PRRSV (porcine reproductive and respiratory syndrome virus). In experimental settings, FLUBVs were able to replicate in the respiratory tracts of guinea pigs and ferrets [23,24]. However, it cannot be ruled out that the host range of FLUBVs is broader than currently understood, or that they will likely expand over time [25].

The important epidemiological feature of concern with FLUBVs (potential emergence of radically new pandemic viruses) is primarily constrained by: the virtual absence of a natural reservoir; and a lack of gene reassortment between human and animal influenza viruses. There is reassortment of genes within different FLUBV lineages. Antigenic drift, due to accumulation of mutations in genes encoding hemagglutinin (HA), neuraminidase (NA), and NS1, also occurs. Overall, these indicate a high level of adaptive evolution [26,27,28,29].

Based on HA antigenic and genetic variation, two distinct FLUBV lineages are identified. These discrete lineages (B/Yamagata/16/88-like (B_Yam_) or B/Victoria/2/87-like (B_Vic_) viruses) have been co-circulating globally since at least 1983 [30,31,32,33,34]. According to Chinese scientists, the two major HA lineages possibly began to diverge in the early 1970s, with the Victoria lineage emerging in China through gradual evolution from a minor lineage [35]. As mentioned, HA features differentiate them. Already by 1988, 27 amino acid (a.a.) residues were different between the HA1 domains of these two lineages [30]. Few a.a. changes were detected by this time between HA2 domains, and a phylogenetic analysis indicated that the HA2 domains of circulating B/Yamagata/16/88-like and B/Victoria/2/87-like viruses were also on separate lineages [31]. However, the HA a.a. sequence homology between the two FLUBV lineages is higher than between the two subtypes of seasonal FLUAV [36]. A full genome analysis shows that Yamagata/Victoria differences are not limited to the HA gene. The PB1 and PB2 phylogenetic trees exhibit a deep divergence similar to the situation with the HA gene. The PB1, PB2, and HA genes were consistently derived from a single lineage [29,37]. Thus, the Yamagata/Victoria distinction is currently restricted to a PB2-PB1-HA complex [29]. Antibody cross-protection between the two B lineages is assumed to be low [4,38], although some studies suggest cross-reactive serum antibodies may be significant [39]. https://www.ncbi.nlm.nih.gov/pmc/articles/PMC6338745/ (accessed on 20 March 2022)—CR16 Under experimental conditions, the B_Yam_ lineage may induce cross-antibody responses to the Victoria lineage, but the opposite is not as efficient [40]. An investigation of B-cell memory and monoclonal antibodies (mAbs), isolated from individuals vaccinated with quadrivalent seasonal vaccine (IIV-4), confirmed the immunological dominance of B/Yamagata HA [41].

After the appearance of the two lineages, B_Vic_ dominated in the late 1980s, while B_Yam_ prevailed in the 1990s. Since 2001, both FLUBV lineages have been co-circulating globally each epidemic season [26]. Furthermore, viruses of both lineages can co-circulate, in different or equal proportions, in the same season and in certain areas [10,42,43]. 

Vijakrishna et al. have shown that, in general, B_Yam_ are is conserved and demonstrates weaker antigenic drift. They are less able to evolve through antigenic drift and are under weaker immune selection pressure compared with B_Vic_. Viruses of the B_Vic_ lineage contain more a.a. substitutions in HA near the receptor binding site (RBS). Changes in B_Yam_ viruses are less frequent and tend to appear in sites more distant from the RBS [29]. Later, using a larger dataset (2651) of FLUBV whole genome sequences and a more comprehensive analytic model, Langat et al. [38] found no significant difference in the antigenic drift rate between the B_Vic_ and B_Yam_ lineages. This supports the view of Bredford [44].

Although the B_Vic_ and B_Yam_ lineages have comparable rates of antigenic drift, their evolutionary dynamics are different. Yamagata-lineage viruses form two co-circulating clades: clade 2 (B/Massachusetts/02/2012) and clade 3 (B/Wisconsin/1/2010) that also segregate genetically across the whole virus genome, while Victoria-lineage viruses show antigenic drift of a single lineage. [45]. B_Yam_ populations have a prolonged absence of intra-lineage reassortment and show alternating dominance between antigenic groups [45]. In contrast, B_Vic_ viruses show evidence of reassortment between clades within the Victoria lineage over time. B_Vic_ persists for longer in local geographic regions before wider dissemination [44].

In general, a phylogenic analysis indicates greater B_Yam_/B_Vic_ inter-lineage reassortment (NA, NP) than Yamagata intra-lineage reassortment [37]. It is interesting that inter-lineage reassortment, in which B_Vic_ viruses acquire B_Yam_ genes, has been more frequent than the reverse. Co-circulation B_Yam_ and B_Vic_ lineage viruses can also undergo reassortment of segments PA, NP, NA/NB, M1/BM2, and NS1/NS2 [29,46,47,48]. 

Most studies show a predominance of the B_Vic_ lineage over the B_Yam_ lineage in younger populations [5,10,49,50,51]. A comprehensive comparative analysis of the epidemiological features of both viral lineages, using the Global Influenza B Surveillance (GIBS) database, also confirmed an uneven age distribution of B_Vic_ and B_Yam_ infections. It was revealed that B_Yam_ cases are, on average, older than B_Vic_ cases. In most countries, B_Vic_ cases tended to be distributed in younger people (0–25 y.o.) along a unimodal curve, with a peak below 10 years of age. The age distribution of B_Yam_ cases frequently followed a bimodal curve: a larger, primary peak (‘those below 10 years of age’); and a smaller, secondary peak (usually ‘those 25 to 50 years of age’) [1]. One possible explanation is different durations of post-infection immunity for B_Vic_ and B_Yam_ viruses. Another reason for uneven age distribution is that viruses of different lineages preferentially bind different sialic acid receptors in human respiratory airways. B_Vic_ viruses appear to have both α-2,3- and α-2,6-linked sialic acid-binding capacities; B_Yam_ viruses predominantly bind α-2,6-linked sialic acid [52]. It is known that α-2,3-linked sialic acid glycans are expressed more highly in the respiratory tissues of children than in those of adults [53].

An interesting trend has been identified using GIBS: an unequal geographical distribution of B_Vic_ and B_Yam_ infections. B_Vic_ was relatively more frequent in tropical countries, while B_Yam_ was more frequent in temperate climate countries [1]. One of the more likely reasons for the uneven geographical distribution of the two FLUBV lineages lies in the diverse demographic structures of countries globally. Countries located around the tropics have a lower median age, on average, than those in temperate climates [1].

## 3. Organization of the Influenza B Virion and Genome. Viral Proteins as Cross-Protective Vaccine Targets

The influenza B virus, like the influenza A virus, belongs to the family *Orthomyxoviridae*. The genomes of both viral types are comprised of eight negative-sense, single-stranded RNA segments encoding 11 proteins [54]. In both viruses, FLUA and FLUB segments 1, 3, 4, and 5 encode just one protein per segment: the PB2, PA, HA and NP proteins, respectively (Figure 2) [55,56]. 

Influenza B viruses feature a few differences. Segment 2 of FLUAV encodes the polymerase subunit PB1. In some strains, this segment also codes for the small accessory protein PB1-F2 with apoptotic activity [57]. The FLUBV genome has no segment encoding this accessory protein. Segment 6 of FLUAV encodes only the NA protein, while that of FLUBV encodes both the NA protein and the NB matrix protein [58]. Segment 7 encodes the M1 matrix protein in both influenza A and B viruses. Segment 7 also encodes, by RNA splicing [59,60], the M2 ion channel in FLUAV. In FLUBV, the BM2 membrane protein is expressed from segment 7 in an alternate (+2) reading frame. BM2 is necessary for the production of viable viral progeny [61,62], and the decreased incorporation of M2 into virions reduces the viral infectivity. Segment 8 encodes the interferon-antagonist NS1 protein and, by mRNA splicing, the NEP/NS2 protein in both FLUAV and FLUBV [55,56]. Generally, the organization of the influenza B virion is similar to the influenza A virion, with four envelope proteins: HA; NA; and BM2 and NB (instead of M2) [54]. Like the FLUAV M2 protein, the BM2 protein is a pH-sensitive proton channel that is essential for the uncoating process [63]. The BM2 protein forms tetramers with polar residues lining the pore that is formed, a feature important for the inactivity of adamantanes against BM2 [9,64]. The NB protein is unique to influenza B viruses. It is incorporated into the virion, and it possesses: an 18-residue N-terminal ectodomain; a 22-residue transmembrane domain; and a 60-residue cytoplasmic tail [65,66]. The NB protein is thought to also have ion channel activity [54]. Unlike the ion channel BM2, however, NB has no pH-modulating activity in the trans-Golgi network (AM2, BM2 and CM2 ion channels do) [67]. It has been shown that this protein is not required for viral replication in vitro but promotes efficient replication in vivo [68]. In infected animals, viruses lacking NB induced lower levels of IFN-ɑ, but titers and body weight were unchanged [69].

Hemagglutinin is the main influenza virus surface glycoprotein. It is responsible for viral attachment and host cell entry via interaction with sialic acid. The functional HA unit is a homotrimer, with each monomer comprised of two domains, HA1 and HA2, linked by a disulfide bond. HA2 forms the membrane anchor and the long alpha-helix ‘stem’ or ‘stalk’, and HA1 forms the distal globular head that contains the receptor binding site (RBS) and the majority of the antigenic sites [70]. The head domain of HA contains four immunodominant, highly variable antigenic sites: the 120 loop; the 150 loop; the 160 loop; and the 190 helix. These are the main targets of neutralizing antibodies and are subject to antigenic drift. Neutralizing antibodies prevent infection by blocking interaction between the RBS (in the head of HA) and sialic acid receptors (on host cells). The blocking of HA attachment to the cell occurs through the interaction between the antibody’s Fab fragment and the HA RBS. 

Other ways to prevent or mitigate an influenza infection are: inhibition of interaction between the virus and endosomal membranes during viral entry into cells; inhibition of viral egress; and Fc receptor-mediated effector functions, such as antibody-dependent cell-mediated cytotoxicity (ADCC) and antibody-dependent cellular phagocytosis (ADCP) [71,72,73]. These Fc–Fcγ receptor interactions are necessary for broadly reactive HA head-, HA stalk-, and NA-directed antibodies to mediate protection in vivo [74]. Broadly neutralizing antibodies (bnAbs) play a key role in these processes. It is noteworthy that, in adult human populations, HI antibody titers induced by the non-canonical sites are almost as high as those induced by the classical sites, and the non-canonical antibody responses appeared to increase with age [75].

## 4. Conserved Protein Epitopes as Target Antigens for the Development of Broadly Protective Influenza B Vaccines

Successful development of broadly protective influenza B vaccines requires information about conserved epitope locations and the mechanisms of action of broadly protective antibodies [76,77]. Many mAbs with broad reactivity against the IBV surface proteins HA and NA have been described in the last decade [76,78,79,80,81]. In vivo protection, against both lineages, has been demonstrated in mice by passive transfer, through non-neutralizing antibody-dependent effector functions [76,82,83,84].

Cross-reactive humoral responses, between IBV lineages in specific contexts (seasonal vaccine strains, primary human infections), have also been described [39,84]. Broad FLUBV recognition and broad prophylactic/therapeutic protection, against FLUBV infection in vivo, can be mediated at alternative epitopes [85]. Broadly protective mAbs bind conserved epitopes localized: in the stalk of the HA; in the residual esterase domain at the base of the HA head; in a specific area of the HA globular head; and in the NA enzymatic site. It is worth noting that broadly protective, HA-targeting mAbs are different with FLUAV and FLUBV. With FLUAVs, the majority of broadly protective mAbs target the stalk domain; only a few exceptional mAbs bind conserved, head domain epitopes. With FLUBVs, broadly protective anti-head antibodies are more common [85]. Some of these mAbs are represented in Table 1.

Dreyfus et al. obtained three broadly neutralizing antibodies using B cells of a vaccinated human that bound HAs from both lineages: CR8033; CR8071; and CR9114. Passive administration with a low dose of these mAbs (separately) fully protected mice against B_Yam_ and B_Vic_ lineage viruses. The CR8033 and CR8071 mAbs neutralized representative viruses of both FLUBV lineages in vitro. The CR9114 mAb did not display neutralizing activity in vitro but provided complete protection against infection in vivo; this suggests participation of the Fc-mediated effector mechanism of antibody action [78]. It is noteworthy that mAb CR9114 also has protective properties against influenza A subtypes H1, H3, H7, and H9. Antibodies CR8033, CR8071, and CR9114 recognized different antigenic HA epitopes. According to a 3D reconstruction, three CR8033 Fabs bind the HA trimer on an epitope overlapping the receptor binding pocket and surrounding antigenic sites. CR8071 binds the vestigial esterase domain at the base of the HA head. CR9114 binds and neutralizes an epitope on the HA stalks of influenza A and B.

A human-derived mAb, 5A7, has the ability to neutralize FLUBVs by binding the C terminus of HA1 in the stalk. Its protective effect against lethal infection with B_Vic_ strains was shown in a mouse model [86]. Three mAbs (34B5, 33F8, 46B8) that neutralized FLUBVs of both lineages and ancestral strains, were obtained by Chai et al. [87]. Antibody 46B8 was the most effective (in neutralization assay) against all tested FLUBV strains spanning over 70 yrs. It binds a conserved epitope in the vestigial esterase domain of HA, and it blocks FLUBV infection by preventing low pH-induced conformational changes in HA during the membrane fusion step. However, it does not block viral attachment to the cell surface receptor. The 46B8 mAb mediated antibody-dependent cytotoxicity, although the contribution of another Fc dependent effector functions (such as an antibody-dependent phagocytosis, antibody-dependent respiratory lysis) to the in vivo protection by 46B8, cannot be excluded.

The C12G6 mAb cross-neutralizes representative viruses spanning 76 years of FLUBV antigenic evolution, including viruses belonging to the B_Yam_, B_Vic_, and earlier lineages [82]. C12G6 features broad prophylactic and therapeutic activity (in mice and ferrets) and has an effect comparable to oseltamivir. Epitope mapping indicated that C12G6 targets a conserved epitope overlapping the receptor binding site in the HA region. This indicates why it neutralizes virus so potently. C12G6 inhibits FLUBV (preventing viral entry, egress, and HA-mediated membrane fusion) via the antibody-dependent cell-mediated cytotoxicity and complement-dependent cytotoxicity responses. 

A number of mAbs to HA, both neutralizing and non-neutralizing, were obtained from human B cells using single-cell screening technology [80]. These mAbs recognized HAs from strains in both lineages. Eleven of them were stalk-binding non-neutralizers, and nine were neutralizers. Four of the latter exhibited activity in HI assay and were not subject to further analysis. The epitopes of four stalk-binding neutralizing mAbs (excluding TRL847) and two previously published FLUBV stalk-binding mAbs 5A7 [88] and CR9114 [78]) were determined using the chemical linkage of peptides onto scaffolds (CLIPS) technology. Convergent patterns of peptide recognition were seen in some cases. For example, the TRL849 mAb and the previously characterized CR9114 mAb recognized a related set of peptides, even though they have different a.a. sequences and exhibit different activities in vivo [80]. Three mAbs (TRL845, TRL848, TRL849) were broadly reactive and neutralizing against a panel of multiple strains from both IBV lineages. They, as well as the published mAb 5A7, demonstrated high therapeutic efficacy in vivo.

A significant proportion of the 22 broadly reactive (mouse) mAbs specific for FLUBV HA, obtained and characterized by Arunkumar et al. [76], recognized and bound a conserved domain on the stalk (on the alpha helix, to be precise). Although these mAbs did not display neutralizing activity in vitro with a wide range of purified FLUVs, they completely or partially protected against a lethal dose of FLUBV from either lineage. A strong correlation, between the level of protection and the activities of the respective mAbs in an Ag-specific ADCC reporter assay (measuring engagement with the Fcγ receptor), was also shown. The authors suggest that protection is mediated by Fc-dependent effector functions, but it seems to be epitope independent [76].

The degree of protection provided by cross-reactive antibodies has a clear hierarchy [41]. For example, it was shown that Abs capable of mediating HIA displayed the greatest prophylactic and therapeutic protection against experimental FLUBV challenge. Monoclonal Abs conferred intermediate protection, characterized by broad FLUBV recognition but no HIA activity. These antibodies provide protection by engagement with host effector cells via Fc receptors. Finally, mAbs binding the IBV stalk domain, failed to neutralize in vitro and provided the weakest protection against experimental challenge. It is noteworthy that, unlike highly strain-specific HIA Abs to FLUAV, anti-FLUBV mAbs elicited by seasonal influenza vaccines generally recognized all strains tested within a respective FLUBV antigenic lineage spanning over 20 years of antigen drift [41]. Isolation of multiple human FLUBV stem-binding Abs can both: neutralize in vitro; and provide potent, cross-lineage protection in mice [80]. In human populations, serum antibodies binding the FLUBV stem are widely prevalent, with titers increasing with age or following IBV infection [88,89]. 

Studies using FLUAV have established that antibody-based protection in the murine challenge model can be mediated via direct neutralization of virus and/or engagement with host effector cells via Fc receptors (FcR) https://www.ncbi.nlm.nih.gov/pmc/articles/PMC6338745/ (accessed on 20 March 2022)—CR35 [74,90,91]. Fc function plays a critical role in protection when an mAb loses its ability to neutralize the virus via its Fab domain. The ability to engage with the FcR protective function is highly dependent on epitope location on the virus. In general, HA stalk-specific antibodies mediated ADCC and displayed cross-reactivity with FLUBV of both phylogenic lineages [84] and across viruses evolving over time [76].

Previously, highly conserved epitopes of the FLUBV NA globular head domain and mAbs directed against the FLUBV NA were detected, although the antibodies were not assessed for in vivo protection, and structures of the antibody bound to NA were not solved [90,92]. It was later shown that NA-reactive mAbs isolated from mice and rabbits have protective effects against both IBV lineages [81]. Since anti-NA Abs are able to bind NA of heterologous strains, thereby inhibiting viral release from infected cells and subsequent viral transmission, they are able to form a broad defense within viral subtypes [81,92,93,94,95].

Using hybridoma technology, Wohlbold et al. [81] identified a panel of five broadly cross-reactive murine mAbs against FLUBV NA. All of the mAbs demonstrated cross-reactivity with purified whole viruses and recombinant NA in enzyme-linked lectin assays (ELLAs) and functionally inhibited NA enzymatic activity in vitro. They protected mice when administered prophylactically at the tested dose (5 mg/kg), followed by challenge with five murine lethal doses (mLD_50_) of FLUBVs belonging to either HA lineage. [81]. The 1F2 mAb exhibited superior efficacy to the standard care of oseltamivir treatment, when administered at 72 hpi in a mouse challenge model.

Conserved NA epitopes are located on the head of the molecule, and they are distinct from the enzymatic active site. Electron microscopic analysis of the complex, between RNA and the Fab portions of antibodies 1F2 and 4F11, has shown that the binding sites of both Fabs appear not to directly overlap the NA enzymatic active site. Thus, direct contact with the catalytic site may not be required for the inhibition of NA. It may instead occur by binding or steric hindrance of substrate access to NA [81]. The structural footprints of 4F11 and 1F2 are adjacent to each other, but separate. All five mAbs displayed ADCC activity when incubated with MDCK cells infected with the B/Malaysia/2506/04 (B_Vic_) or B/Florida/04/06 (B_Yam_) viral strains. 

The a.a. residues of the binding footprints are highly conserved across all FLUBVs, which is consistent with the broad binding profiles of the 4F11 and 1F2 mAbs [81]. Neuraminidase-binding mAbs may be able to increase the level of ADCC achieved by HA stalk-binding mAbs during natural infection, most likely by providing increased contact points for Fc–Fc receptor engagement on the surface of the virus or infected cells [81,96]. 

A detailed analysis of the FLUBV NA B cell response in humans indicates concurrent expansion of NA-specific peripheral blood plasmablasts 7 days after IIV immunization [79]. These plasmablasts express mAbs with antiviral activity against FLUBVs of both lineages (B_Vic_, B_Yam_) and feature prophylactic/therapeutic activity in mice. These FLUBV NA-specific B cell clonal lineages persisted in CD138^+^ long-lived bone marrow plasma cells. This investigation confirmed the ability of IIV to induce a subpopulation of FLUBV NA-specific B cells with broad protective potential, which is important for the development of broadly protective influenza vaccine [79].

However, such broadly cross-reactive Abs are rare and immunosubdominant compared to strain-specific Abs to the variable HA head. Conversely, broadly cross-reactive CD8^+^ T cells are abundant and can account for substantial immune responses to influenza [97]. T-cell immunity correlates with a broad protection against influenza infections [98,99]. CD8^+^ T cells (CTLs) make a special contribution to a broad range of heterosubtypic protective immunity [95,100,101]. Influenza-specific CD8^+^ T cells provide cross-protection across different IAV subtypes [102,103] and both FLUBV lineages [104]. CD4^+^ T cells play an important, often central role, in the immune response providing ‘help’ to B cells in the synthesis of effective neutralizing antibodies by increasing affinity and switching antibody classes. They also promote the development and maintenance of a virus-specific CD8^+^ T lymphocyte response [105]. In addition, CD4^+^ T cells targeted to highly conserved influenza virus proteins are cross-reactive and provide protection against even novel and potentially pandemic strains of influenza [106]. CD4^+^ effector T cells provide IFNγ-independent protection, both by stimulating B cell maturation and antibody production, and via perforin-mediated cytolytic activity (which provides control of viral replication) [107,108]. Such non-traditional mechanisms of action of CD4^+^ T cells should be taken into account when developing vaccines aimed at providing effective T-cell immunity [109]. 

T cells do not provide neutralizing immunity against influenza viruses, but reduce disease severity and duration of infection, thereby facilitating recovery from illness [98]. A number of CD4^+^ and CD8^+^ T-cell epitopes are highly conserved in internal proteins such as NP, M1, and PB1 [110]. Such conserved epitopes have become one of the major strategies for the development of T cell-based cross-protective influenza vaccines.

Three FLUBV NP epitopes have been identified for CD8^+^ T cells [101,111]. One of them, NP_82–94_ (MVVKLGEFYNQMM), is restricted by HLA-A*021. The other two (NP_30–38_ RPIIRPAT, NP_263–271_ ADRGLLRDI) are restricted by HLA-B8. However, only the NP_30–38_ peptide was able to induce interferon gamma production from FLUBV-specific polyclonal CTLs [104]. It is assumed that CTL responses against FLUBV are preferentially directed against HLA-B8 epitopes [112]. The above HLAs are common in the human population; this suggests that NP T-cell epitopes are expected to be recognized by a majority of individuals owing to the high prevalence of this allele globally.

Preexisting PB1_413–421_^+^ CD8^+^ memory T cells have been detected in the blood and lung tissues of healthy donors, with clonal expansion upon infection with FLUAV or FLUBV. In addition, such CD8^+^ T cells were found in the majority (80%) of tested donors, and thus are abundant across HLA-A*02:01^+^ donors [97]. The mechanism of protection with vaccine candidate BM2SR is partly driven by a response towards more universal CD8^+^ T-cell epitopes found in the FLUBV HA2 stalk [113]. The CD8^+^ epitope YYSTAASSL (FLUBV HA2_190_) has a high affinity for MHC class I and effectively induces the production of IL-2 and TNF-α in mouse splenocytes. Thus, determining the cross-reactivity of T cells against FLUAV and FLUBV is a key step in understanding universal, anti-influenza T-cell immunity. Such cross reactivities raise the prospect of designing a T cell-based UIV. 

## 5. Strategies for Developing Cross-Protective Influenza B Vaccines

A number of alternative strategies are being investigated for their potential to serve as a universal influenza vaccine (UIV) platform. Delivery platforms include: replicating or non-replicating viral vectors; recombinant VLPs; recombinant protein or peptide vaccines; and RNA/DNA vaccines. Further, UIV development efforts have mainly focused on FLUAVs. However, the development of vaccines against both influenza types (A/B) is the main goal in the field of influenza vaccine development [114].

To date, two universal vaccines that include FLUAV and FLUBV antigens have been developed, and they are in clinical trials only (Table 2). A new vaccine, Multimeric-001, was designed to protect against seasonal and pandemic influenza virus strains, regardless of mutation. It contains conserved, linear epitopes from specific proteins (HA, NP, M1) of influenza virus types A and B [115]. The vaccine induces both humoral and cellular immunity [116,117]. The cell-mediated immunity and elevations in HAI levels support M-001’s potential as a universal vaccine. The vaccine has another important property: M-001 priming resulted in enhanced seroconversion towards all three strains of trivalent influenza vaccine (TIV), compared to priming with a placebo [117].

FLU-v is another example of a “universal” influenza vaccine candidate that may provide long-lasting protection against most influenza A/B strains. The vaccine contains four polypeptides representing immunoreactive, conserved regions within the influenza virus: M1, 32 a.a. (DLEALMEWLKTRPILSPLTKGILGFVFTLTVP); NPA, 21 a.a. (DLIFLARSALILRGSVAHKSC); NPB, 20 a.a. (PGIADIEDLTLLARSMVVVR); and M2, 24 a.a. (IIGILHLILWILDRLFFKCIYRLF). FLU-v was administered subcutaneously with adjuvant ISA-51 [113]. The adjuvant is composed of a light mineral oil and a surfactant system designed to make an oil/water emulsion. The chosen subcutaneous route combines ease of delivery and increased exposure to dermal antigen-presenting cells. Vaccine-induced CD8^+^ T-cell responses can, in the absence of neutralizing Abs, protect mice against a lethal challenge with influenza virus [118]. As demonstrated in clinical studies, cellular immune responses to FLU-v correlated with reductions in viral shedding and symptoms after influenza challenge (A/H3N2, A/H1N1) in volunteers [119]. To date, FLU-v has successfully completed a phase IIb clinical trial [120]. 

Another universal vaccine model (influenza A/B) was developed at Jilin University [121]. Researchers used the norovirus (NoV) P protein as a platform for the presentation of conserved epitopes of HA from influenza viruses A/H1N1, A/H3N2, and B. The NoV P protein is capable of self-assembling into virus-like particles and is considered a good platform for vaccine development against infectious diseases [122]. The virus-like P particle is formed by 24 copies of the protrusion (P) domain of the NoV capsid protein; it is easily expressed and purified, extremely stable, and highly immunogenic [122,123]. Each P domain contains three surface loops, which have been demonstrated to be useful for foreign antigen presentation. Consensus HA2_90–105_ peptide sequences for FLUAV H1, FLUAV H3, and FLUBV were inserted into loops 1, 2, and 3, respectively. More precisely: one copy of the H1 FLUAV consensus HA2_90–105_ sequence (DIWTYNAELLVLLENE) was inserted into loop 1 (between residues G274 and T275); one copy of the H3 FLUAV consensus HA2_90–105_ sequence (DLWSYNAELLVALENQ) was inserted into loop 2 (between residues S372 and N373); and one copy of the FLUBV consensus HA2_90–105_ sequence (DTISSQIELAVLLSNE) was inserted into loop 3 (between residues G392 and S393) of the NoV P domain via a GGGGS linker.

After the immunization of mice, this chimeric P particle induced a strong and specific IgG Ab response against subtype-specific HA2 epitopes. Vaccination with trivalent HA2-PP significantly reduced viral titers in the lung after challenge. It is interesting that serum Ab titers induced by trivalent HA2-PP were boosted by a subtype-specific virus, but not *vice versa*.

Several conserved viral antigens of HA stalk, NA, and internal protein (PB1, NP, M1) T-cell epitopes have been defined as targets for eliciting cross-reactive immune responses [124,125]. The first broadly protective vaccine candidate against influenza B was created by a group of researchers working in a commercial Merck laboratory in 2004 [126]. The vaccine features a genetic fusion between: a cleavage site conserved among all FLUBVs; and the membrane surface protein complex of the bacterium *Neisseria meningitidis*. Mice immunized with a recombinant protein including this sequence showed 100% survival after lethal FLUBV challenge from either lineage. In addition, this sequence (PAKLLKER GFFGAIAGFLE) was not only the same for FLUBVs in both phylogenetic lineages, but also had a high degree of homology with the corresponding FLUAV site. At the end of 2017, the first phase of clinical trials of this peptide vaccine began [114].

Previous studies have shown that Abs targeting the HA stalk of FLUAV can be broadly protective (156). Broadly protective vaccines against both influenza A subtypes are being successfully developed using the strategy of chimeric HA (cHA) molecules [127,128]. A vaccination scheme based on chimeric HAs has been adapted to FLUBV. The vaccine strain’s HA contains head domains derived from exotic FLUAV HA subtypes (H5, H7, or H8) and stalk domains from IBV HAs [129]. The constructs were subcloned into mammalian expression vectors, then transfected into human embryonic kidney 293T (HEK 293T) cells. Chimeric viruses were also expressed in the baculovirus system to create antigen proteins for vaccination. To reorient the immune response from the immunodominant sites of the HA head toward the conserved HA stalk domain, mice were first primed with plasmid DNA expressing cH5/B, followed by protein vaccinations with cH7/B and cH8/B antigens. The chimeric HA/B vaccine regimen induced cross-reactive, non-neutralizing Abs against a wider range of diverse FLUIBVs and protected mice against lethal challenge with FLUBVs from both phylogenic lineages. The results from serum transfer experiments and antibody-dependent cell-mediated cytotoxicity (ADCC) assays indicate that this protection is mediated by Abs, likely via Fc effector functions. 

Later, a novel approach to universal influenza B vaccine development, based on “mosaic” HA (mHA), was described [130]. Mosaic HAs were constructed by replacing four major antigenic sites of the virus B/Yamagata/16/88 with the corresponding sequences from different exotic influenza A HAs (H5, H8, H11, H13). The replacement allows: overcoming the dominance of the major antigenic sites; and redirection of the immune response to the subdominant, conserved epitopes in the head and stalk domains. Mosaic HAs were expressed using the baculovirus/insect cell expression system [75,131]. A vaccination regimen included a plasmid DNA (mH13/B_Yam_) prime, followed by two boosts with a different mHA at 3-week intervals. As expected, the immunization with candidate influenza B vaccine (B mHAs) induced broadly reactive Ab responses toward B viruses of both phylogenic lineages. A sequential challenge with 5 LD_50_ of influenza B viruses showed that vaccination with mHA resulted in protection against homologous and heterologous FLUBVs. The authors suggested that viruses with mHA induce Abs with a stronger ADCC activity than the viruses with B cHA. This was explained as follows: the B mHA approach is able to elicit a different Ab repertoire; it contains cross-reactive, non-HI active, and non-neutralizing head Abs that induce stronger ADCC than the anti-stalk Abs. Moreover, B mHAs have an advantage relative to those of B cHAs. This is because recombinant viruses with B mHAs grow robustly in eggs and would be suitable for multiple vaccine production platforms, including recombinant HA, inactivated, and live attenuated constructs.

A high level of protection in an animal model was shown by a novel, M2-deficient, single-replication (M2SR) influenza B vaccine [132]. The RNA of the vaccine virus includes: segments 1, 2, 3, 5, 8; segment 7 lacking the entire BM2 open reading frame (ORF) from influenza B/Lee/1940; and the HA and NA vRNA (segments 4, 6) from influenza B/Brisbane/60/2008 (B/Bris60, B_Vic_ lineage) or B/Wisconsin/01/2010 (B/WI01, B_Yam_ lineage). The M2SR vaccine virus (lacking the coding sequence for the BM2 ion channel protein) is replication-deficient in normal MDCK cells, but grows in complementing MDCK cells that stably express the BM2 protein.

Both the BM2SR-WI01 and BM2SR-Bris60the vaccines are non-pathogenic in vivo, and BM2SR viruses do not replicate in the murine respiratory tract. The vaccines induced moderate cellular infiltration in the lungs and stimulated only a limited inflammatory response. They induced serum and mucosal Abs (IgA, IgG) as well as cellular responses against both FLUBV lineages. Each BM2SR vaccine provided complete protection against infection with drifted and heterologous FLUBV in an in vivo lethal model. The cross-protection is most likely provided by cross-reactive Abs directed against the highly conserved stalk region of the HA. In 2019, this vaccine completed phase II clinical trials.

Traditional vaccines focus on the formation of Abs to the main surface antigen of the influenza virus, HA, and little attention is paid to neuraminidase as an antigen for influenza vaccines. As a result, seasonal influenza vaccines, unlike influenza virus infection, poorly represent key NA epitopes and rarely induce NA-reactive B cells [42,95,133]. There is evidence, however, that modern quadrivalent vaccines (QIVs) induce NA Abs with broad, potent antiviral activity against both lineages in humans [95]. Indeed, NA Abs can provide reliable, broad protection and could potentially be elicited prophylactically with new vaccine strategies.

Immunization with recombinant NA of influenza virus B/Yamagata/16/88 adjuvanted with 5 µg of poly(I ⋅ C) protected mice from challenge with 1.1 × 10^6^ PFU of the homologous B_Yam_ virus or the heterologous FLUBV strains Vic87 and Mal04, both of which belong to the antigenically distinct Victoria lineage [79]. Influenza B virus NA has not diverged into two lineages like FLUBV HA, which may partially explain the good cross-reactivity. Moreover, conserved linear B-cell epitopes have been detected in FLUBV NA [134]. One of them (epitope DILLKFSPTEITAPT) has high-affinity binding with some well-known MHC class II alleles and can be used as a target antigen for a cross-protective FLUBV vaccine.

It has been shown that recombinant NA protein immunization can significantly contribute to the reduction of FLUBV shedding and prevents or limits virus transmission from guinea pigs infected with either B_Yam_ or B_Vic_ FLUBVs to naïve recipients [93]. However, viral titers in animals receiving i.m. or i.n. vaccination were similar, and only i.n. vaccination blocked transmission effectively. The authors suggest mechanisms of decreased viral shedding. One is that even a slight reduction in nasal titers can adversely impact transmissibility of influenza viruses. A second explanation is that Abs inhibiting the enzymatic activity of viral NA alter virus transmissibility. It has been reported that in the absence of NA activity, influenza viruses tend to aggregate [98], which may negatively impact transmissibility. In addition, the virus might be trapped by decoy receptors on natural defense proteins such as mucin, also leading to aggregation and reduced transmission ability. 

Finally, it is possible that the virus is efficiently released from an infected host and transmitted to a susceptible one, but that the virus is coated by anti-NA Abs that impair its ability to initiate a new infection, thus becoming trapped by mucins in the respiratory tract of the recipient. The significant role of mucosal immunity against viral NA, and less importantly serum Ab, in preventing efficient inter-host transmission has been shown by other investigators [135,136].

Various viruses are used as vectors for vaccine target antigens. One of the effective vaccine platforms for inducing both mucosal and systemic immunity, and providing broad protection, are adenovirus vector vaccines incorporating conserved influenza virus antigens [137,138,139]. The targeting of highly conserved T-cell antigens has become one of the major strategies for the development of cross-protective influenza vaccines. Previously, it was shown that only a single dose of i.n. vaccination with rAd-encoding NP protected mice from infection with heterologous FLUAVs, including the H1N1 pandemic strain [140]. A Korean researcher obtained replication-defective adenoviruses (rAd) encoding highly conserved NP (derived from B/Yamagata/16/1988 or B/Shangdong/7/1997) and developed a universal vaccine candidate [141,142]. The NP of the B/Yamagata/16/1988 virus includes a dominant CTL epitope, FSPIRITFL (rAd/B-NP(Y), while B/Shangdong/7/1997 (Victoria lineage) includes epitope FSPIRVTFL (rAd/B-NP(V) with one a.a. variation in B-NP position 171. Hence, these two strains carry either isoleucine or valine, respectively. BALB/c mice were vaccinated i.n. with the two designs and examined for NP-specific immune responses. Both rAd/B-NP vaccines are equally immunogenic and induce similar NP-specific CD8^+^ T-cell and NP-specific humoral immune responses. Significant and similar numbers of Dd/NP(Y) 166–174 or Dd/NP(V)166–174 tetramer-specific CD8^+^ T cells were seen despite different viral challenges in identically vaccinated mice. Mice immunized with the vaccine survived homologous/heterologous FLUBV challenge and showed little or no morbidity. Similar data were obtained by Dhakal et al. [143]. After A/NP-r Ad or B/NP-r Ad vaccination, mice demonstrated robust systemic and pulmonary vaccine-specific B-cell and T-cell responses. Vaccine candidates generated long lasting protection against diverse influenza strains. Protection with NP-based vaccines correlates with the establishment of resident memory CD8^+^ T cells in lungs. Therefore, NP could be further developed as a cross-protective vaccine for FLUBV or as one of the two components of a universal FLUAV/FLUBV vaccine. 

There has long been concern that potent, local T-cell responses might damage the lungs, but it has been experimentally proven that, despite CD8^+^ T-cell responses in the lungs, the lungs were not damaged and functioned normally after vaccination [142,143]. There were no differences in total lung capacity, lung compliance, pulmonary resistance to airflow, or pulmonary diffusion capacity permitting gas exchange between PBS-inoculated mice and mice vaccinated with either A/NP-rAd or B/NP-rAd [143]. This study provides important support for vaccines based on T cell-mediated protection.

A promising and interesting strategy is engineering chimeric A/B viruses that can provide cross-type immunity. Several research groups have obtained various chimeric constructs. The engineered reassortants between influenza A and B viruses were obtained using PR/8/34, A/WSN/33, A/Len/134/17/57, or B/Yamagata/16/88 as the viral backbone [144,145,146,147,148]. Immunogenic and protective properties of some of these engineered reassortants have been investigated in animals; they have shown various degrees of protection against challenge by different types of viruses [147,148].

## 6. Conclusions

Until 2020, influenza B was one of the most common and severe respiratory infections, with a trend towards increasing incidence. The emergence of SARS-CoV-2 has changed the etiological structure of acute respiratory infections. However, it is obvious that, in accordance with the principles underlying epidemic process dynamics, SARS-CoV-2 will not dominate as aggressively in 2–3 years, and the problem of preventing other acute respiratory viral infections, including influenza B, will continue to be relevant. 

Although FLUBV antigenic drift is less pronounced than that of FLUAV, seasonal vaccines need to change their FLUB vaccine strain every to 2–4 years. Simultaneous circulation of viruses belonging to different phylogenetic lineages and to two antigenically different B_Yam_ clades complicates vaccine strain choices and highlights the clear need to develop a broadly protective vaccine. In choosing a designated FLUB vaccine strain, another relatively more difficult task (compared with FLUA), is regional endemic strain variation. This can lead to situations in which the WHO-recommended vaccine strain does not match those circulating locally. 

The described broadly protective mAbs indicate that conserved epitopes are present in virus surface glycoproteins. They, like conserved epitopes of internal proteins common to viruses of both phylogenetic lineages, can be the basis for the development of broadly protective influenza B vaccines. Combining broadly neutralizing Abs with abundant cross-reactive CD8^+^ T cells is important for optimal universal protection against distinct influenza strains. Such vaccines induce immune responses from both the cellular and humoral branches of the immune system, thereby providing long-term protection. 

In parallel with the creation of a universal FLUA/FLUB vaccine, FLUB-only vaccines are being developed. The importance of the latter lies not only in FLUB prevention, but also in the very attractive prospect of eradication of FLUBV from the human population [88,94]. The lack of a sustained animal reservoir, as well as the relatively small phylogenetic divergence of FLUBV glycoproteins, make the development of a broadly protective vaccine a real possibility in the near future.

## Figures and Tables

**Figure 1 viruses-14-01323-f001:**
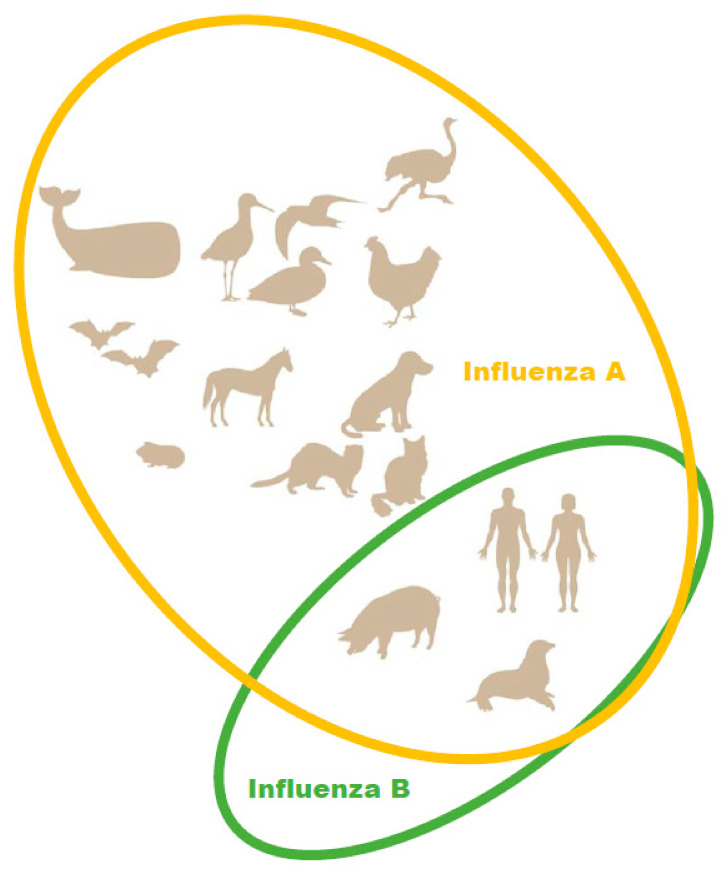
Natural hosts of influenza A and B viruses.

**Figure 2 viruses-14-01323-f002:**
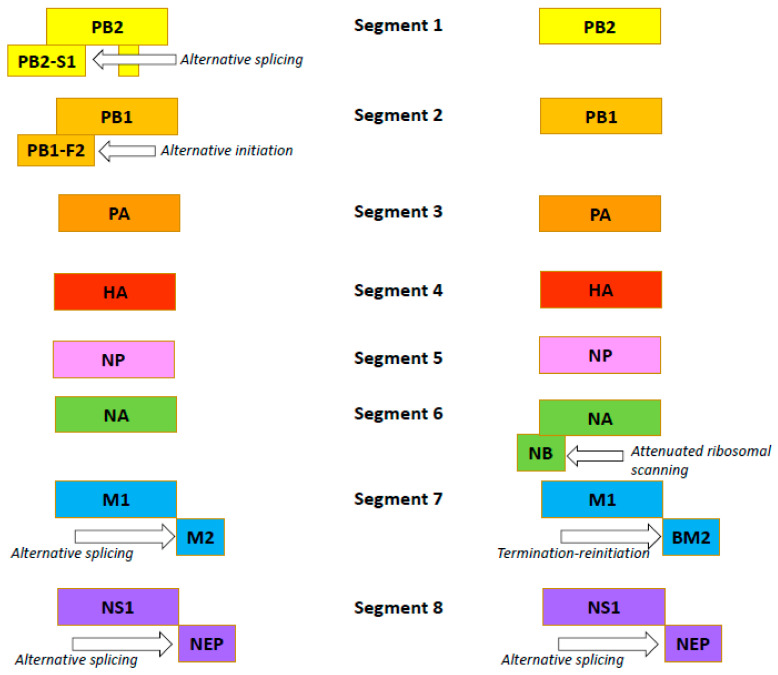
Organization of the influenza A (the left side) and B (the right side) viruses’ genomes.

**Table 1 viruses-14-01323-t001:** Cross-reactive monoclonal antibodies against influenza B virus.

mAb (Origin)	Target Protein	Action	References
CR8033 (h)	HA head, epitopes overlapping receptor binding pocket and surrounding antigenic sites	Blocks viral attachment. Binds and neutralizes both IBV lineages and has neutralizing activity against B_Yam_ strains	[78]
CR8071 (h)	HA head, the vestigial esterase domain at the base of the HA head	Binds to the vestigial esterase domain, neutralizing activity in vitro, intermediate protection in vivo	
CR9114 (h)	HA stalk, epitope conserved across influenza A and B viruses	Blocks the pH-induced conformational changes in HA and membrane fusion, non-neutralizing in vitro, high protection against diverse IBV in vivo	
5A7 (h)	C terminus of HA1 in the stalk	Blocks viral attachment and membrane fusion, low relative neutralizing potency	[86]
46B8 (h IgG1)	HA, vestigial esterase domain at the base of the HA head	Blocks HA-mediated membrane fusion by preventing low pH-induced conformational changes	[87]
C12G6 (m/h)	HA, conserved epitopes in close proximity to the receptor binding pocket (RBP)	Prevents viral entry, viral egress. Displayed ADCC, neutralizing activity against B_Yam_, B_Vic_ viruses. Binds a similar epitope to CR8033, overlapping the receptor binding domain	[82]
TRL784, TRL799, TRL811, TRL812- TLR813, TRL835, TRL837, TRL841, TRL842, TRL846, TRL856 (h)	HA stalk domain	No neutralizing activity	[80]
TRL845, TRL847, TRL848, TRL849, TRL854 (h)	HA stalk domain	Neutralizes diverse strains of both IBV lineages	
KL-BHA-6D12 **, KL-BHA-2C6 *, KL-BHA-2G4 *, KL-BHA-4C10 **, KL-BHA-8G3 ** (m)	HA stalk domain, long alpha-helix	No neutralizing activity in vitro Completely (**) or partially (*) protects against a lethal IBV dose from either lineage, effects through the ADCC, ADCP	[76]
KL-BHA 8G12 **, KL-BHA-4G12 * (m)	HA stalk domain, outside alpha-helix		
KL-BHA-1B5 **, KL-BHA-2H11 ** (m)	Globular head domain		
KL-BHA-1D2 **, KL-BHA-2H10 **, KL-BHA-9C6 **, KL-BHA-3A10 **, KL-BHA-3H10 *, KL-BHA-8A5 ** (m)	Globular head domain or conformational epitopes of stalk		
R95–1E07, R95–1D05, K77–2D09 K77–2D11 (h)	HA head, epitopes proximal to RBP	Bind and neutralize both IBV lineages and have HI activity. Cross-reactive Abs capable of mediating HI showed the greatest protective effect in vivo	[41]
R95–1F04, R95–1C01, R95–1E05 (h)	HA head domain	Conferred intermediate protection, characterized by broad IBV recognition, but no HIA activity. Protection mediated by mAb engagement with cellular Fc receptors An absence of neutralizing activity in vitro and provided the weakest protection against experimental challenge Bind Fc receptor	
W85–3F06, R95–1E03, R95–2A08	HA stalk domain		
W85–1A07, W85–3E10 (h)	Within the vestigial esterase domain at the base of the HA		
1F2(m) 1F4(m)	Surface of the NA tetramer, not directly overlapping the NA enzymatic active site	Enhance viral clearance, display ADCC activity	[81]
3G1(m)	NA, epitope overlaps or adjacent to the enzymatic active site		
4B2 (m)	Surface of the NA tetramer, right above both the 1F2 and the 4F11 footprints	Enhances viral clearance, displays ADCC activity	
4F11 (m)	Surface of the NA tetramer, not directly overlapping the NA enzymatic active site	Enhances viral clearance, displays ADCC activity	
1086C12, 1086F8, 1092D4, 1092E10, 11122C6 (h)	NA	Inhibits NA enzymatic activity and blocks the release of progeny virions	[79]
HCA 2	NA		[88]

**Table 2 viruses-14-01323-t002:** Influenza B universal vaccine candidates.

Candidate Vaccine or Target Antigens	Vaccine Platform	Approach	Stage of Development	Developer, Partners	References
Multimeric- 001(M-001)	Recombinant protein	Recombinant protein featuring conserved epitopes of M1, NP, HA (FLUAV), and M1, NP (FLUBV)	Clinical trials (Ph III, 2020)	Biond Vax Pharmaceuticals (Israel)	[115,116,117]
FLU-V	Peptide-based	Construct derived from conserved regions of internal proteins (M1, FLUAV-NP, FLUBV-NP, M2)	Clinical trials (Ph IIb, 2020)	Imutex Pep Tcell (SEEK) (UK), EndFluenza (UK)	[118,119,120]
BM2SR	M2-deficient single-replication live virus	M2-deficient single-replication vaccine for influenza B virus	Clinical trial (Ph II, 2019)	FluGenInc, USA, The Biomedical Research Institute (CA, USA), University of Tokyo (Japan)	[132]
Chimeric HA	Chimeric virus	Recombinant HA including head domains (FLUAV H5, H7, or H8) and FLUBV stalk domains	Preclinical (2021)	Icahn School of Medicine at Mount Sinai (USA)	[129]
Mosaic HA	Recombinant virus	HAs were constructed by replacing four major IBV antigenic sites with the corresponding sequences from different FLUAV HAs (H5, H8, H11, or H13)	Preclinical (2021)	Icahn School of Medicine at Mount Sinai (USA)	[130]
rNA proteins	Recombinant protein	Influenza B virus NA from Yam88 adjuvanted with 5 µg of poly(I C)	Preclinical (2021)	Icahn School of Medicine at Mount Sinai (USA)	[94]
rAd/B-NP(Y) and rAd/B-NP(V)	Viral vector	Replication-defective adenoviruses (rAd) encoding the conserved NP epitopes FSPIRITFL (B_Yam_) or FSPIRVTFL (B_Vic_)	Preclinical (2019)	Ewha Womans University, Seoul (South Korea)	[141,149]
HA2 aa90–105	Nanoparticle	FLUAV (H1, H3) and FLUBV HA subtype consensus HA2_90–105_ peptides were inserted into loops 1, 2, and 3 of norovirus P protein, respectively.	In development	Jilin University (China)	[121]
A/NP-rAd B/NP-rAd	Viral vector	Replication-deficient (E1 and E3 deleted) adenovirus-5 (rAd) vectors expressing NP antigens from A/PR/8/34 or B/Ann Arbor/1/86	In developmentt	Johns Hopkins University Bloomberg School of Public Health, Baltimore, MD (USA)	[142]

## Data Availability

Not applicable.

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
