# Peer review of "Influenza B: Prospects for the Development of Cross-Protective Vaccines"

_viruses, 2022, doi:10.3390/v14061323_

Round 1

Author Response

Response to Reviewer 1 Comments

Dear reviewer, we are grateful to you for careful reading of the manuscript and the comments you made.

Point 1:      Repetition of content in the abstract and introduction “The circulation of two antigenically different phylogenetic…” For reviews, abstract should not rewrite any of the content in the introduction or discussion, rather say “what is being discussed and covered in the article”.

Response 1: Abstract is corrected according your recommendation

Point 2:      I recommend authors to provide information on the CD4 cells helps in boosting neutralizing cross reactive antibody response.

Response 2: The information about role of CD4+ in anti-virus immunity is inserted to p.16

Reviewer 2 Report

The review by Liudmila et al. explores the different aspects of influenza B, focusing on strategies for developing cross-protective vaccines. Influenza B has always been a neglected agent due to the relevance of Influenza A; however, the importance of influenza B has been growing recently. Influenza B can cause disease with equal or higher consequences than influenza A. Therefore, a review article about influenza B and vaccines is completely within the interest of the virology community. The authors provide a good review of the literature on basic concepts of epidemiology, differences between lineages, and genome organization. Authors later focus on strategies for the development of cross-protection. Although the literature review on the topics covered is good, information is missing about strategies under pre-clinical evaluation. Furthermore, live attenuated influenza vaccines are entirely omitted, which is a topic that must be covered in any vaccine review for influenza. Finally, the lack of figures in the review must be addressed before the study is suitable for publication.

Major comments

Figures should be generated that summarize the information provided. A review article should have a summary of information where readers can get that information through the use of figures. The lack of figures in this article will significantly negatively impact the metrics since readers always focus first on the figures of a review article. At least 2 different figures should be generated. A review without figures is not up to standards, and the current version of this review should not be accepted if this is not appropriately addressed. 

The manuscript requires the incorporation of promising strategies under pre-clinical development. One entirely omitted strategy in the review was live attenuated influenza vaccines. These strategies always will be considered one of the most promising ones due to their similarities to a natural infection. Several reports are available regarding the development of different LAIV using mutations like the current FDA-approved platforms, genome rearrangements, and the use of more recent strains carrying the modifications in the B/Ann Arbor/1/86. Information encompassing the current state of the art of LAIV for FLUBV should be incorporated.

Please modify the text so is clear when the information provided about different strategies is from samples collected from humans or animal models. More context is the information provided will facilitate the interpretation of the information

Minor comments

Update influenza B acronyms based on the most recent guidelines (IBV to FLUBV). Same where influenza A is mentioned

Please revisit the information about the higher number of infections in young populations. It is slightly contradictory with the information about more than 50% fatality due to FLUBV.

33/636. Is this number calculated based on all respiratory infections of influenza (A+B) infections?

..’’ It is widely believed that influenza B is a less severe infection than influenza A, but current studies challenge this notion. There is no difference between influenza A and influenza B in the frequency of hospitalization, intensive care unit (ICU) admission, or rate of death among hospitalized influenza patients [13]…’ Please rephrase this sentence and add a reference that supports the statements. The term widely believed is ambiguous. 

Using these species as animal models or pets, replication in guinea pigs and ferrets is under experimental conditions using these species as animal models or pets?. Please clarify since both species are well-known animal models to study influenza B

Please correct the following sentence: …’’ Antibody cross-protection between the two B lineages is assumed to be low’’… It is not assumed; it has been demonstrated (Are you referring to neutralizing antibodies?). Provide references that support this

Please provide information about recent patterns of circulation and the sharp decrease of Yam isolates.

Rephase the first portion of the section about the organization of the genome following the segment order to describe the features, similarities, and differences 

Author Response

 Response to Reviewer 2 Comments

Dear reviewer, thank you for attention and careful reading of our manuscript.

Point 1: Figures should be generated that summarize the information provided. A review article should have a summary of information where readers can get that information through the use of figures. The lack of figures in this article will significantly negatively impact the metrics since readers always focus first on the figures of a review article. At least 2 different figures should be generated. A review without figures is not up to standards, and the current version of this review should not be accepted if this is not appropriately addressed.

Response 1: The figures were generated  and inserted on the p.3 and p.6.

Point 2: The manuscript requires the incorporation of promising strategies under pre-clinical development. One entirely omitted strategy in the review was live attenuated influenza vaccines. These strategies always will be considered one of the most promising ones due to their similarities to a natural infection. Several reports are available regarding the development of different LAIV using mutations like the current FDA-approved platforms, genome rearrangements, and the use of more recent strains carrying the modifications in the B/Ann Arbor/1/86. Information encompassing the current state of the art of LAIV for FLUBV should be incorporated.

Response 2:

Indeed, the development of live cross-protective vaccines is a very interesting and important topic. It deserves a separate review. In our review, we have included only brief information on the page 22.

Point 3: Please modify the text so is clear when the information provided about different strategies is from samples collected from humans or animal models. More context is the information provided will facilitate the interpretation of the information.

Response 3: The origin of monoclonal antibodies is indicated in table. 1. Stages of vaccine development (clinical or preclinical trials) are shown in table 2.

Point 4: Update influenza B acronyms based on the most recent guidelines (IBV to FLUBV). Same where influenza A is mentioned

Response 4: Acronyms have been corrected

Point 5: Please revisit the information about the higher number of infections in young populations. It is slightly contradictory with the information about more than 50% fatality due to FLUBV.

Response 5: The noted high mortality from influenza B was in certain epidemic seasons among people 60 aged and over. Young children have a higher incidence of influenza B than the general population. The text contains relevant links.

Point 6: 33/636. Is this number calculated based on all respiratory infections of influenza (A+B) infections?

Response 6: Сlarification is included in the text (page 3) 

Point 7: ..’’ It is widely believed that influenza B is a less severe infection than influenza A, but current studies challenge this notion. There is no difference between influenza A and influenza B in the frequency of hospitalization, intensive care unit (ICU) admission, or rate of death among hospitalized influenza patients [13]…’ Please rephrase this sentence and add a reference that supports the statements. The term widely believed is ambiguous.

Response 7: The data of Su S. et al [13] is cited in this fragment of manuscript

Point 8: Using these species as animal models or pets, replication in guinea pigs and ferrets is under experimental conditions using these species as animal models or pets?. Please clarify since both species are well-known animal models to study influenza B

Response 8: In the text of the manuscript, guinea pigs and ferrets are clearly identified as laboratory animal models.

Point 9: Please correct the following sentence: …’’ Antibody cross-protection between the two B lineages is assumed to be low’’… It is not assumed; it has been demonstrated (Are you referring to neutralizing antibodies?). Provide references that support this

Response 9: Corrected

Point 10: Please provide information about recent patterns of circulation and the sharp decrease of Yam isolates.

Response 10: Viruses influenza B of each leanage circulate with different intensity in different periods of time. In our opinion, information about the decrease in the circulation of influenza B/Yam in recent years is not of principle importance for the topic of this review.

Point 11: Rephase the first portion of the section about the organization of the genome following the segment order to describe the features, similarities, and differences

Response 11: Corrected

Reviewer 3 Report

The authors have comprehensively reviewed the current state of influenza B epidemiology and vaccine development. The authors have also provided practical solutions for further vaccine development, which adds a significant value to the literature.

As the authors have noted, the interest on influenza B have been significantly smaller compared to that of influenza A. Therefore, more attention would arise by articles like this one.

Minor points:

1. In the abstract: VLP, abbreviations may not be recommended in the abstract.

2. Some of the numbering of citations in the manuscripts (multiple citations) are not in increasing orders.

(e.g. [14 , 11, 15, 16], [107, 105, 85, 108, 111] ...)

Author Response

Response to Reviewer 3 Comments

Dear reviewer, thank you for attention and careful reading of our manuscript.

Point 1: In the abstract: VLP, abbreviations may not be recommended in the abstract.

Response 1: Abstract has been corrected

Point 2: Some of the numbering of citations in the manuscripts (multiple citations) are not in increasing orders. (e.g. [14 , 11, 15, 16], [107, 105, 85, 108, 111] ...)

Response 2: The numbering of citations has been corrected